# Flowering and Morphogenesis of Kalanchoe in Response to Quality and Intensity of Night Interruption Light

**DOI:** 10.3390/plants8040090

**Published:** 2019-04-04

**Authors:** Dong Il Kang, Hai Kyoung Jeong, Yoo Gyeong Park, Byoung Ryong Jeong

**Affiliations:** 1Division of Applied Life Science (BK21 Plus Program), Graduate School, Gyeongsang National University, Jinju 52828, Korea; kin01@gnu.ac.kr (D.I.K.); jhksmile@naver.com (H.K.J.); 2Institute of Agriculture & Life Science, Gyeongsang National University, Jinju 52828, Korea; iuyiuy09@gnu.ac.kr; 3Research Institute of Life Science, Gyeongsang National University, Jinju 52828, Korea

**Keywords:** light intensity, light quality, light spectrum, night break, photomorphogenesis

## Abstract

The effects of the quality and intensity of night interruption light (NIL) on the flowering and morphogenesis of kalanchoe (*Kalanchoe blossfeldiana*) ‘Lipstick’ and ‘Spain’ were investigated. Plants were raised in a closed-type plant factory under 250 μmol·m^−2^·s^−1^ PPFD white light emitting diodes (LEDs) with additional light treatments. These treatments were designated long day (LD, 16 h light, 8 h dark), short day (SD, 8 h light, 16 h dark), and SD with a 4 h night interruption (NI). The NIL was constructed from 10 μmol·m^−2^·s^−1^ or 20 μmol·m^−2^·s^−1^ PPFD blue (NI-B), red (NI-R), white (NI-W), or blue and white (NI-BW) LEDs. In ‘Spain’, the SPAD value, area and thickness of leaves and plant height increased in the NI treatment as compared to the SD treatment. In ‘Lipstick’, most morphogenetic characteristics in the NI treatment showed no significant difference to those in the SD treatment. For both cultivars, plants in SD were significantly shorter than those in other treatments. The flowering of Kalanchoe ‘Lipstick’ was not affected by the NIL quality, while Kalanchoe ‘Spain’ flowered when grown in SD and 10 μmol·m^−2^·s^−1^ PPFD NI-B. These results suggest that the NIL quality and intensity affect the morphogenesis and flowering of kalanchoe, and that different cultivars are affected differently. There is a need to further assess the effects of the NIL quality and intensity on the morphogenesis and flowering of short-day plants for practical NIL applications.

## 1. Introduction

Light drives photosynthesis in plants, helping them to build carbon-based materials, and further acts as an environmental signal; plants respond to the intensity, wavelength, duration and direction of light [1]. Light is the primary energy source for plants and the most important regulatory factor in the life cycle of a plant: it affects the seed germination, seedling establishment, transition to flowering, and morphogenesis [2,3].

Night interruption (NI) during short day (SD) seasons effectively accelerated the flowering of long day plants (LDPs) and allowed earlier marketing or seed production, and delayed the flowering of short day plants (SDPs) during long day (LD) seasons [4]. NI for 4 h has usually been applied to both SDPs and LDPs in commercial greenhouse production [5]. The NI using low-intensity light emitting diodes (LEDs) is supposed to slightly affect the net photosynthesis and ensure growth promotion in tomatoes [6]. NI has effectively accelerated the growth and development of herbaceous LDPs such as *Campanula carpatica* [7], *Coreopsis grandiflorum* [8], and *Cyclamen persicum* [9] in SD seasons. By inducing vegetative growth, NI has also been used to prevent or delay the flowering of herbaceous short-day plants (SDPs) such as *Dahlia pinnata* [10] and *Dendranthema grandiflorum* [11]. Low-intensity NI (3–5 μmol·m^−2^·s^−1^ PPFD) was used to effectively promote flowering of juvenile *Cymbidium aloifolium* with increased growth rates [12].

NI with red (R) light was shown to be the most effective in inhibiting flowering, and far-red (FR) light was able to revert this inhibitory effect in many cases [13]. As the FR light proportion increased (as the R:FR ratio decreased), flowering was increasingly promoted in *Petunia hybrida* (LDP) and inhibited in *Antirrhinum majus* (SDP) [14]. The NI with FR light delayed flowering of the day neutral plant *Pelargonium × hortorum* [15]. The NI with a combination of R and blue (B) lights (R:B = 1:1) more effectively promoted the flowering of *Cyclamen persicum* (LDP) than NI with B, R, or FR light alone [16]. For *D. grandiflorum*, NI with R light most effectively inhibited flowering, while NI with B or FR light wasn’t very effective [17]. The NI with a combination of R and B lights (R:B = 1:1), and R light alone effectively inhibited the flowering of *D. grandiflorum* [18]. In SDPs, low-intensity NI with B light, or B light supplemented with R and FR lights did not affect the flowering nor growth attributes [10,19]. In a combined shifting treatment of B and W light, the NI ending with B light (NI-WB) induced flowering [19]. Thus, it is probable that the B light receptor can enhance floral-inducer activity, resulting in a higher energy requirement for the promotion of flowering [19].

There has been little published on the effects of NI with LEDs on the morphogenesis and flowering, and especially the effects of NI on Crassulacean acid metabolism (CAM) plants. Furthermore, as LEDs have the benefits of easily controlled duration, intensity and quality, new practical applications of NI with LEDs for floricultural crop production are needed. In this study, the effects of the NIL quality and intensity on the morphogenesis and flowering of SDPs, kalanchoe (*Kalanchoe blossfeldiana*) ‘Lipstick’ and ‘Spain’, were examined.

## 2. Materials and Methods

### 2.1. Plant Materials and Growth Conditions

Cuttings of kalanchoe (*Kalanchoe blossfeldiana* ‘Lipstick’ and ‘Spain’), a qualitative short day plant (SDP), were grown on a commercial kalanchoe farm (J Flower, Gimhae, Korea) and stuck in 50-cell plug trays that contained a 4:1 mixture (v/v) of a commercial medium (Bio No.1 medium, Heungnong Co., Incheon, Korea) and perlite (Pergreen, No. 1, G-Biotech Co., Ltd., Yeongi, Korea) on January 25th, 2018. After 55 days, rooted cuttings were transferred to the same closed-type plant factory (7700 cm long × 2500 cm wide × 2695 cm high, Green Industry Co. Ltd., Changwon, Korea) maintained at 20 °C with a relative humidity of 60 ± 10%. The plants of kalanchoe ‘Lipstick’ and ‘Spain’ (respectively about 18.8 cm and 19.3 cm in height) were grown for 98 and 71 days, respectively, at a distance of 20 cm under the NILs. After planting, the plants were fertigated with a multipurpose greenhouse nutrient solution (737.0 mg·L^−1^ Ca(NO_3_)2·4H_2_O, 343.4 mg·L^−1^ KNO_3_, 163.2 mg·L^−1^ KH_2_PO_4_, 43.5 mg·L^−1^ K_2_SO_4_, 246.0 mg·L^−1^ MgSO_4_·H_2_O, 80.0 mg·L^−1^ NH_4_NO_3_, 15.0 mg·L^−1^ Fe-EDTA, 1.40 mg·L^−1^ H_3_BO_3_, 0.12 mg·L^−1^ NaMoO_4_·2H_2_O, 2.10 mg·L^−1^ MnSO_4_·4H_2_O, and 0.44 mg·L^−1^ ZnSO_4_·7H_2_O).

### 2.2. Photoperiodic Light Treatments

Long day (LD, 16 h light, 8 h dark), short day (SD, 8 h light, 16 h dark), and SD with 4-h NI conditions were constructed with 250 ± 10 μmol·m^−2^·s^−1^ PPFD white LEDs (PCL-D9WWCN10SC; Powerlightec Co. Ltd., Suwon, Korea) to grow the plants in a closed-type plant factory (Figure 1). 

The critical day length required to induce flowering of the SDP used throughout this study was 12 h, and hence, an uninterrupted dark period longer than 12 h was sufficient to initiate flowering. However, SD with 16 h of darkness were used in this study to confirm a clear flowering response. Here, 10 μmol·m^−2^·s^−1^ or 20 μmol·m^−2^·s^−1^ PPFD LEDs were used to provide NI with either blue (450 nm, NI-B), red (660 nm, NI-R), white (~400–800 nm, NI-W), or blue and white (450 nm, ~400–800 nm, NI-BW) lights from 23:00 to 03:00 every day.

The average PPFD of each treatment was measured with a digital photometer (HD2102.1, Delta OHM, Padova, Italy) 20 cm above the bench top, and was adjusted to be the same before NI treatments were initiated. A spectroradiometer (USB 2000 Fiber Optic Spectrometer, Ocean Optics Inc., Dunedin, FL, USA) was used to scan the spectral distribution 25 cm above the bench top in 1 nm wavelength intervals. For each light treatment, the spectral distribution and characteristics were measured at three different locations within the plant growing areas.

### 2.3. Data Collection and Statistical Analysis

#### 2.3.1. Growth Characteristics

Growth parameters, such as plant height, number of leaves per plant, the length and width of the third leaf from the top, fresh and dry shoot and root weights, days after treatment to visible flower bud initiation or days for visible buds (DVB), percentage of plants flowered, and number of flowers per plant were measured at 98 and 71 days, respectively, after initiating the photoperiodic treatments in kalanchoe ‘Lipstick’ and ‘Spain’. The number of leaves was established as the number of leaves, excluding cotyledons, that were longer than 1 cm. An electronic scale (EW 220-3NM, Kern and Sohn GmbH., Balingen, Germany) was used to measure the fresh and dry weights. Divided samples of shoots and roots were dried for 72 h at 70 °C in a drying oven (Venticell-222, MMM Medcenter Einrichtungen GmbH., Munich, Germany) before the dry weight of shoots and roots was measured. A leaf area meter (LI-3000, LI-COR Inc., Lincoln, NE, USA) was used to measure the leaf area.

#### 2.3.2. SPAD Value and Chlorophyll Estimation

A chlorophyll meter (SPAD-502, Konica Minolta Inc., Osaka, Japan) measured the SPAD value of the third uppermost matured leaf, prior to harvesting the samples. Three measurements were taken for each sample. The 10 mg fresh samples were taken from young, fully developed leaves to estimate the chlorophyll concentration, where 80% cold acetone was used to extract chlorophyll. The absorbance of the supernatant was measured with a spectrophotometer (Biochrom Libra S22, Biochrom Co. Ltd., Holliston, MA, USA) at 663 nm and 645 nm after centrifugation at 3000 rpm. Calculations were done according to Dere et al. [20].

#### 2.3.3. Harvesting, Microtomization, and Staining

At 53 days after initiating the photoperiodic treatments, stems were harvested from the apex to the third node from plants grown in SD, LD, or different NI treatments. The apex segments were fixed with 2.5% glutaraldehyde according to the method of Karnovsky [21]. After fixation, the plant materials were embedded in paraffin, and cross-sectioned with a microtome (RM2235, Leica Microsystems Ltd., Northvale, NJ, USA) to 7 μm sections. The sections were stained with hematoxylin and photographed using an optical or confocal microscope (BX61VS, Olympus, Hamberg, Germany).

#### 2.3.4. Statistical Analysis

This experiment utilized a randomized complete block design constructed of 3 replications and 3 plants per replication. To minimize the effects of photoperiodic treatment positioning, the lighting locations were randomly mixed between replications in a controlled environment. A statistical analysis program (SAS 9.1, SAS Institute Inc., Cary, NC, USA) was used for statistical analysis. Data from this experiment were subjected to Tukey’s range test as well as an analysis of variance (ANOVA). Graphs were generated with SigmaPlot (SigmaPlot 12.0, Systat Software Inc., San Jose, CA, USA). 

## 3. Results

### 3.1. Morphogenesis

In this study, the plant height, leaf length, and leaf width were the greatest for plants in SD (Figure 2A, Figure 3A,B). Additionally, the higher the light intensity of NI, the shorter the plant height, except in the NI-W treatment of ‘Spain’ (Figure 2A). While the leaf area and the number of leaves per plant showed different results (Figure 3C,D), the specific leaf area, SPAD value, and chlorophyll content showed a similar tendency to that of plant height (Figure 4B–D). The shoot fresh weight, root fresh weight, and root dry weight were not significantly influenced by the quality and intensity of the NIL (Table 1).

Kalanchoe ‘Spain’ was not significantly affected by the NIL quality and intensity, and the plant height was the greatest when grown in SD (Figure 2B). The length and width of leaves displayed a similar tendency to that of plant height (Figure 3E,F). The area of leaves grown under R light was significantly greater than the area of leaves grown in other treatments (Figure 3H). The number of leaves was higher for plants in LD and NI treatments than for those in SD (Figure 3G). In this study, the leaf thickness tended to be higher for plants in the flowering-inducing SD and NI-B treatments (Figure 4E). It is thought that there is a correlation between flowering and the leaf thickness. The fresh and dry weights of the shoot were affected by the quality and intensity of the NIL, although there were no definite tendencies (Table 2).

### 3.2. Flowering

In ‘Lipstick’, flowering was not observed in any photoperiodic treatments, but the SD treatment (Figure 5A). Flowering was suppressed in kalanchoe ‘Lipstick’ regardless of the NIL intensity and quality. Flower bud development was slower in the NI treatments than in SD (Figure 5A). Most of kalanchoe ‘Spain’ flower buds were differentiated or fully developed in 10 μmol·m^−2^·s^−1^ PPFD NI-B, while flower buds of kalanchoe ‘Lipstick’ were not differentiated at all (Figure 6).

In ‘Spain’, flowering was observed in plants in the SD and flower buds were differentiated in plants in the 10 μmol·m^−2^·s^−1^ PPFD NI-B (Figure 5B). For plants in 20 μmol·m^−2^·s^−1^ PPFD NI treatments, no flower buds were differentiated regardless of the light quality (Figure 5B). Plants grown in 10 μmol·m^−2^·s^−1^ PPFD NI-B had flower buds differentiated in a distinctly different pattern than flower buds that did not differentiate in the other treatments (Figure 7). In addition, the DVB was significantly shorter for plants in SD than that of plants in 10 μmol·m^−2^·s^−1^ PPFD NI-B. Plants in 10 μmol·m^−2^·s^−1^ PPFD NI-B had differentiated flower buds, but only 70% of them fully developed, and the DVB was 70 days.

## 4. Discussion

### 4.1. Morphogenesis

*Kalanchoe blossfeldiana* is a facultative Crassulacean acid metabolism (CAM) plant [22]. The species employs C3 photosynthesis, and switches to CAM photosynthesis upon aging, flower induction, or the perception of certain environmental stimuli. CAM-inducing stimuli include water and heat stresses, photoperiodicity (SDs which also induce flowering), and the exogenous application of abscisic acid [23]. CAM plants open their stomata at night to permit the passage of atmospheric CO_2_ and close them during the day to retain water. Similar results were reported in a previous study by Noh et al. [24]. The effects of the NIL quality on the morphogenesis of kalanchoe were not as pronounced as they were for chrysanthemum [17].

According to Rustin and Queiroz-Clare [25], the amount of malates produced nightly in CAM plant leaves increases exponentially with the number of SDs the plants are exposed to. These results are similar to the effects of R LED lighting on the growth of lettuce reported by Johkan et al. [26] and Shimizu et al. [27]. The R light treatments increasing the leaf area is reported for several plants, such as wasabi, perilla, and pea [28,29,30]. Brulfert et al. [31] reported that CAM plants continued producing new leaves in LD. The same results were observed in this study. Leaf thickening was associated with extra cell layers or longer palisade cells, which can increase the area-based photosynthetic capacity [32,33,34]. Leaf thickness was reported to be inversely related to day length [35].

### 4.2. Flowering

Different cultivars of kalanchoe are thought to have different sensitivities to B light. The different growth responses to light of plants may be in accordance with the biochemical properties of photoreceptors [36,37,38,39]. The *PhyA* plays a role in the photoperiodic perception [40,41] and promotes flowering in *Arabidopsis* [42], whereas it delays flowering under LD conditions and promotes flowering under SD conditions in rice, especially in the absence of *phyB* [43]. In response to R light, *phyB* delays flowering and promotes de-etiolation and seed germination [44]. Phytochromes are the main photoreceptors that regulate different developmental processes in plants [45]. Therefore, the role of phytochromes and phototropins varies depending on the crop and the photoperiod. Further studies on the role of phytochromes and photortropins in kalanchoe are necessary.

Zaccai and Edri [46] distinguished three stages of *Eustoma grandiflorum* (Raf.) Shinn. ‘Heidi Deep Blue’ flower bud development, and Islam et al. [47] distinguished seven stages of flower development in *Eustoma grandiflorum* (Raf.) Shinn. ‘Fuji Deep Blue’. However, the stages of flower bud differentiation in kalanchoe are not yet well known. Further research is needed to fully understand the flower bud differentiation process of kalanchoe.

## 5. Conclusions

In conclusion, for both cultivars, plants in the SD were significantly shorter than those in other treatments. The flowering of ‘Lipstick’ was not affected by the NIL quality, while ‘Spain’ flowered when grown in the SD and 10 μmol·m^−2^·s^−1^ PPFD NI-B. These results suggest that the quality and intensity of the NIL affect the morphogenesis and flowering of kalanchoe, and that the responses were cultivar-dependent. Further research is needed to determine how morphogenesis and flowering are affected by the NIL quality and intensity in other kalanchoe cultivars. For the control of flowering in crops, more practical considerations should be given to the NI strategy.

## Figures and Tables

**Figure 1 plants-08-00090-f001:**
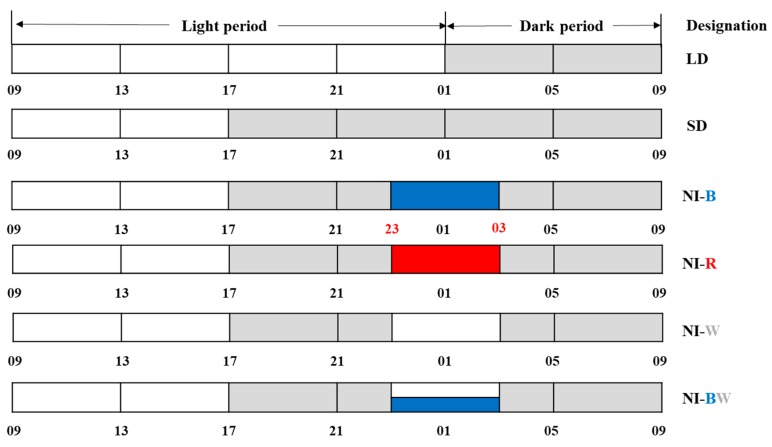
Night interruption (NI) treatments, provided by light emitting diodes (LEDs), used in this study and the night interruption light (NIL) qualities used for 4 h a day during the dark period: NI-B, blue; NI-R, red; NI-W, white; and NI-BW, blue and white. The LD and SD indicate 16 h long day and 8 h short day, respectively.

**Figure 2 plants-08-00090-f002:**
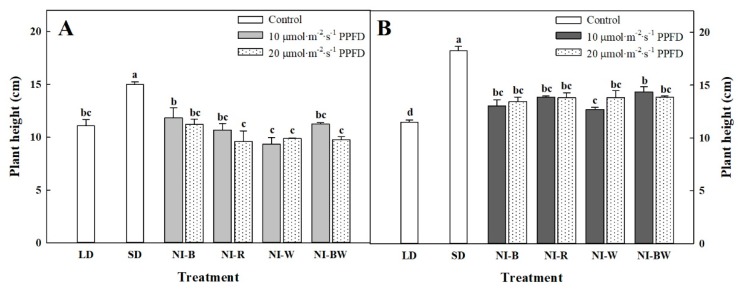
The effects of the night interruption light (NIL) quality at 10 or 20 µmol·m^−2^·s^−1^ PPFD on height of kalanchoe (*Kalanchoe blossfeldiana*) ‘Lipstick’ (**A**) and ‘Spain’ (**B**) measured 98 and 71 days after treatment initiation, respectively. Please refer to Figure 1 for details on the NIL quality. Vertical bars indicate the means ± S.E. (n = 3). Values followed by different letters are significantly different by the Duncan’s multiple range test at *p* ≤ 0.05.

**Figure 3 plants-08-00090-f003:**
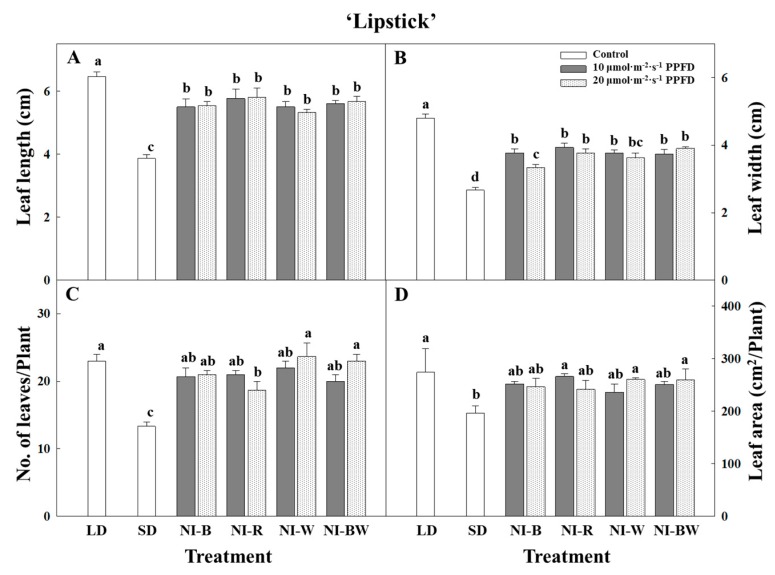
The effects of the night interruption light (NIL) quality at 10 or 20 µmol·m^−2^·s^−1^ PPFD on the leaf length (**A**,**E**), leaf width (**B**,**F**), number of leaves per plant (**C**,**G**), and leaf area (**D**,**H**) of kalanchoe (*Kalanchoe blossfeldiana*) ‘Lipstick’ and ‘Spain’ measured 98 and 71 days after treatment initiation, respectively. Please refer to Figure 1 for details on the NIL quality. Vertical bars indicate the means ± S.E. (n = 3). Values followed by different letters are significantly different by the Duncan’s multiple range test at *p* ≤ 0.05.

**Figure 4 plants-08-00090-f004:**
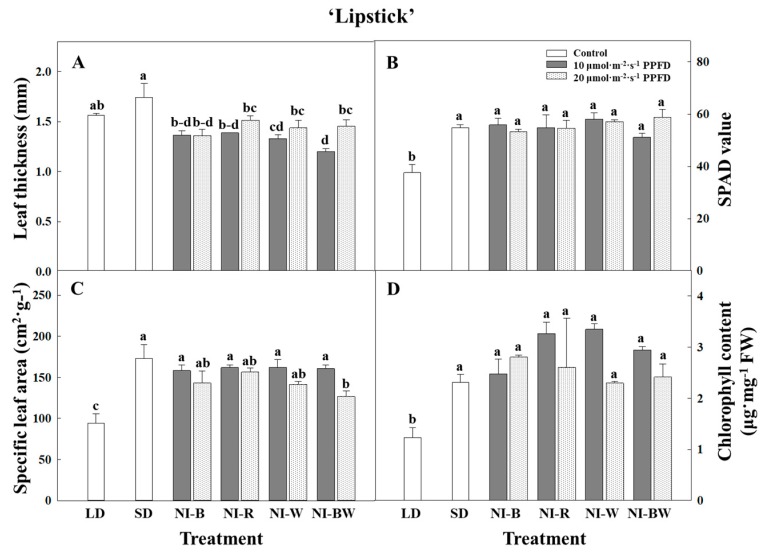
The effects of the night interruption light (NIL) quality at 10 or 20 µmol·m^−2^·s^−1^ PPFD on the leaf thickness (**A**,**E**), SPAD value (**B**,**F**), specific leaf area (**C**,**G**), and chlorophyll content (**D**,**H**) of kalanchoe (*Kalanchoe blossfeldiana*) ‘Lipstick’ and ‘Spain’ measured 98 and 71 days after treatment initiation, respectively. Please refer to Figure 1 for details on the NIL quality. Vertical bars indicate the means ± S.E. (n = 3). Values followed by different letters are significantly different by the Duncan’s multiple range test at *p* ≤ 0.05.

**Figure 5 plants-08-00090-f005:**
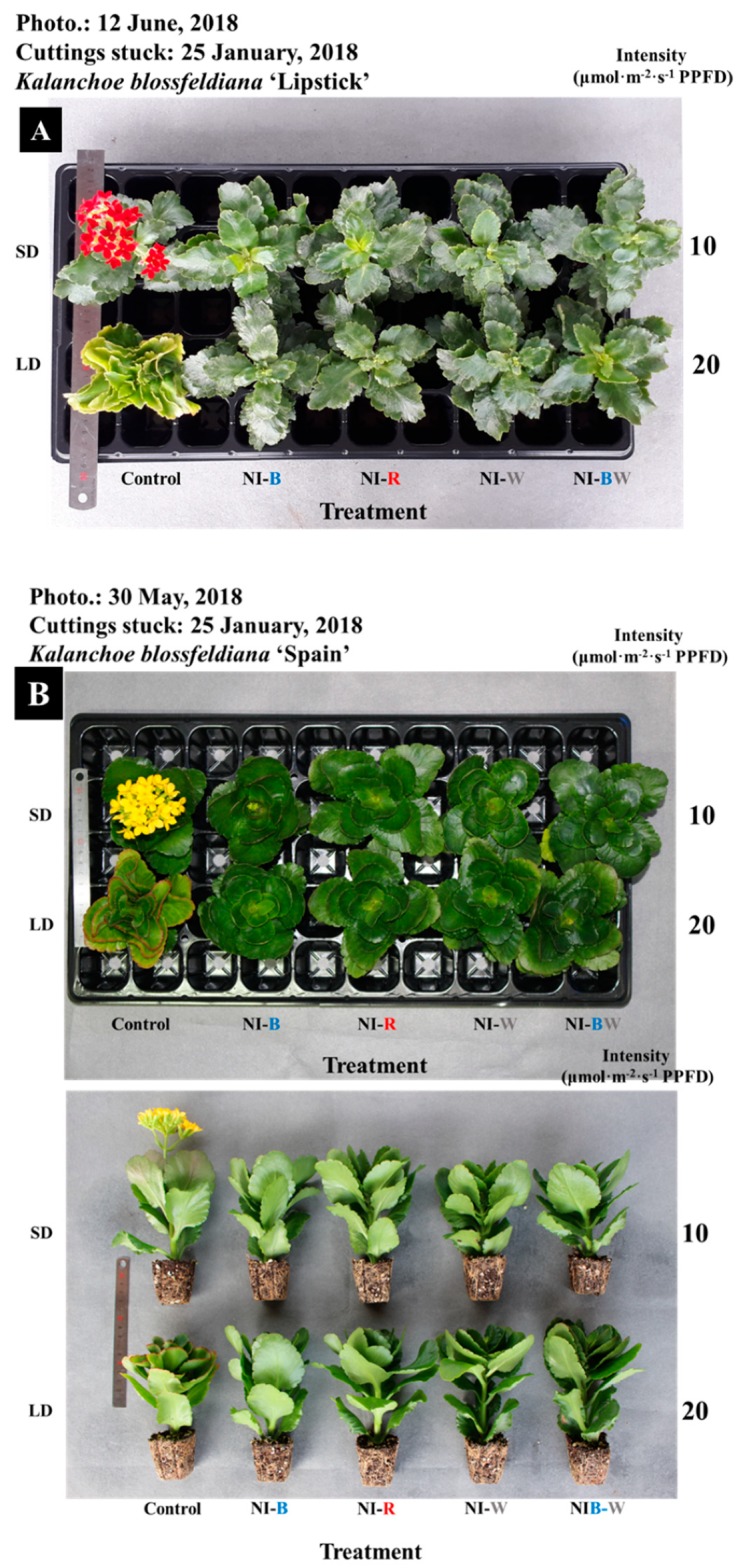
The effects of the night interruption light (NIL) quality at 10 or 20 µmol·m^−2^·s^−1^ PPFD on the flowering of kalanchoe (*Kalanchoe blossfeldiana*) ‘Lipstick’ (**A**) and ‘Spain’ (**B**) measured 98 and 71 days after treatment initiation, respectively. Please refer to Figure 1 for details on the NIL quality.

**Figure 6 plants-08-00090-f006:**
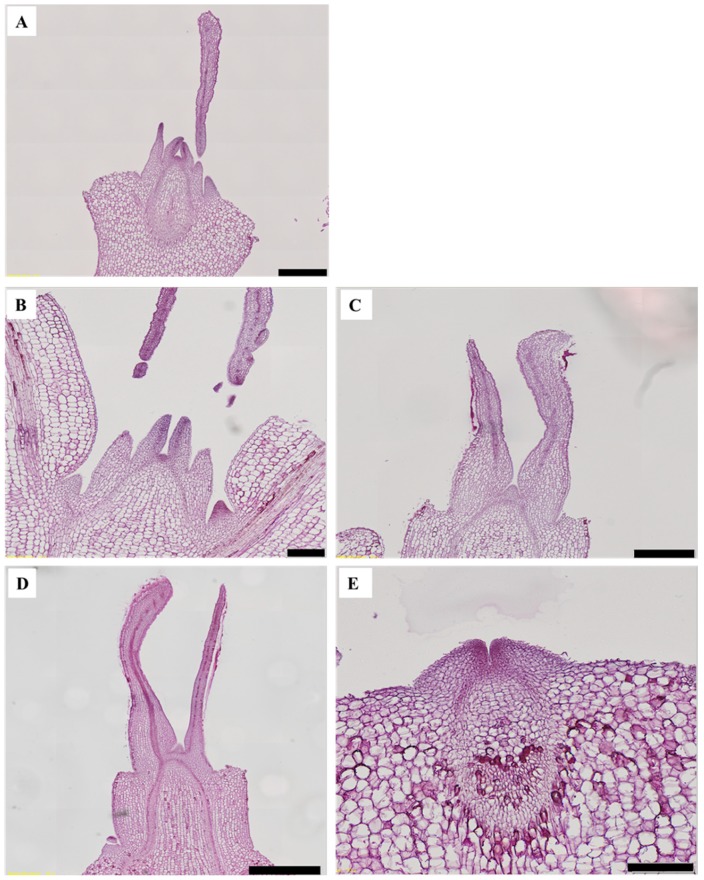
The vertical section of the terminal buds of kalanchoe (*Kalanchoe blossfeldiana*) ‘Lipstick’ grown under NI-B at 10 (**A**) or 20 (**B**) µmol·m^−2^·s^−1^ PPFD; NI-R (**C**), NI-W (**D**), and NI-BW (**E**) at 20 µmol·m^−2^·s^−1^ PPFD for 53 days. Please refer to Figure 1 for details on the NIL quality. Scale bars in A to E indicate 500, 200, 500, 1000, and 200 µm. respectively.

**Figure 7 plants-08-00090-f007:**
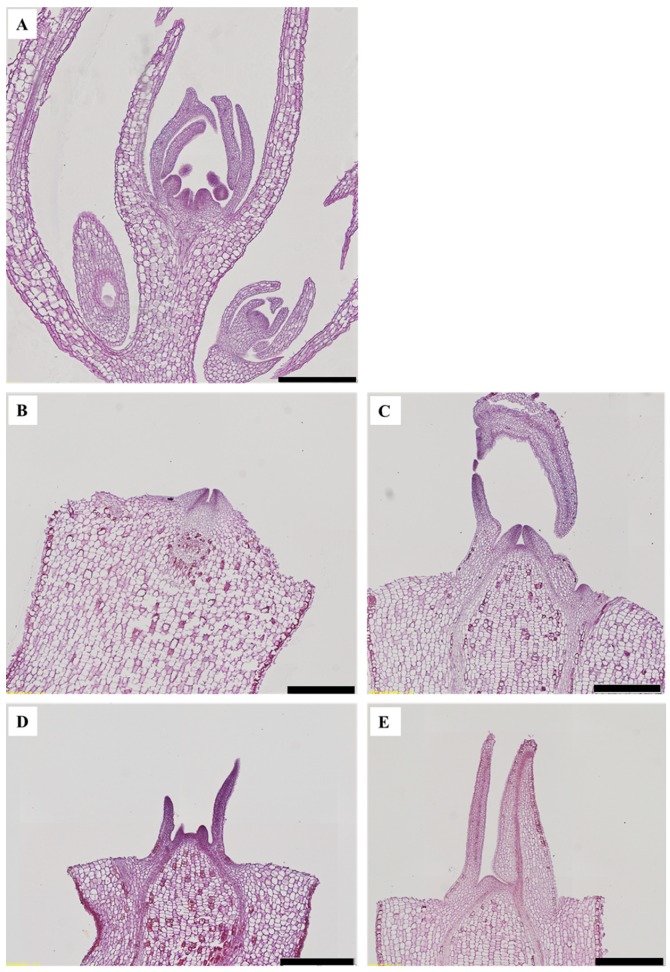
The vertical section of the terminal buds of kalanchoe (*Kalanchoe blossfeldiana*) ‘Spain’ grown under NI-B at 10 (**A**) or 20 (**B**) µmol·m^−2^·s^−1^ PPFD; NI-R (**C**), NI-W (**D**), and NI-BW (**E**) at 20 µmol·m^−2^·s^−1^ PPFD for 53 days. Please refer to Figure 1 for details on the NIL quality. Scale bar indicate 500 µm.

**Table 1 plants-08-00090-t001:** The effect of the night interruption light (NIL) quality at 10 or 20 µmol·m^−2^·s^−1^ PPFD on the fresh and dry weights of kalanchoe (*Kalanchoe blossfeldiana*) ‘Lipstick’ shoot and root measured 98 days after treatment initiation.

Photoperiod	Light Quality ^z^ (A)	Light Intensity (µmol·m^−2^·s^−1^ PPFD) (B)	Fresh Weight (g)		Dry Weight (g)	
			Shoot	Root	Shoot	Root
LD			30.1 ± 1.7	3.3 ± 0.3	3.48 ± 0.2 a ^y^	0.35 ± 0.1
SD			29.7 ± 1.7	2.5 ± 0.1	1.85 ± 0.1 cd	0.26 ± 0.0
NI	B	10	27.1 ± 0.5	2.4 ± 0.2	1.95 ± 0.1 b-d	0.27 ± 0.0
		20	26.9 ± 1.7	2.0 ± 0.2	2.19 ± 0.1 b	0.25 ± 0.0
	R	10	27.9 ± 0.5	2.4 ± 0.2	2.02 ± 0.1 b-d	0.24 ± 0.0
		20	25.8 ± 1.8	2.1 ± 0.2	1.92 ± 0.2 cd	0.24 ± 0.0
	W	10	25.7 ± 2.0	2.0 ± 0.2	1.85 ± 0.2 d	0.20 ± 0.0
		20	28.2 ± 0.8	2.0 ± 0.2	2.28 ± 0.1 bc	0.20 ± 0.0
	BW	10	27.0 ± 0.7	2.2 ± 0.2	1.91 ± 0.1 b-d	0.25 ± 0.0
		20	28.3 ± 2.0	2.5 ± 0.2	2.48 ± 0.1 b	0.50 ± 0.2
F-test ^x^	Photoperiod		NS	NS	**	NS
	Quality (A)		NS	NS	NS	NS
	Intensity (B)		NS	NS	**	NS
	A × B		NS	NS	NS	NS

^z^ Refer to Figure 1 for details on the NIL quality. ^y^ Mean (n = 3) separation within columns by Duncan’s multiple range test at 5% level. ^x^ NS, **: Non-significant or significant at *p* ≤ 0.01.

**Table 2 plants-08-00090-t002:** The effect of the night interruption light (NIL) quality at 10 µmol·m^−2^·s^−1^ or 20 µmol·m^−2^·s^−1^ PPFD on the fresh and dry weights of kalanchoe (*Kalanchoe blossfeldiana*) ‘Spain’ shoot and root measured 71 days after treatment initiation.

Photoperiod	Light Quality ^z^ (A)	Light Intensity (µmol·m^−2^·s^−1^ PPFD) (B)	Fresh Weight (g)		Dry Weight (g)	
			Shoot	Root	Shoot	Root
LD			33.2 ± 1.3 b ^y^	2.9 ± 0.1 b	3.18 ± 0.1 a	0.29 ± 0.0 b
SD			42.7 ± 0.6 a	3.3 ± 0.3 b	2.17 ± 0.0 c-f	0.26 ± 0.0 bc
NI	B	10	33.1 ± 1.4 bc	3.0 ± 0.2 b	1.98 ± 0.1 c-f	0.26 ± 0.0 b
		20	29.9 ± 0.7 cd	2.4 ± 0.3 b	2.41 ± 0.1 b	0.44 ± 0.0 a
	R	10	33.1 ± 1.2 b	3.9 ± 0.8 b	2.40 ± 0.2 b-e	0.30 ± 0.1 c
		20	35.3 ± 1.2 b	4.5 ± 0.2 a	2.28 ± 0.1 b-d	0.26 ± 0.0 bc
	W	10	29.6 ± 0.9 d	2.2 ± 0.4 c	1.99 ± 0.1 f	0.21 ± 0.0 d
		20	33.0 ± 0.4 bc	3.0 ± 0.3 b	1.88 ± 0.0 d-f	0.22 ± 0.0 bc
	BW	10	31.9 ± 1.0 bc	2.5 ± 0.1 b	2.37 ± 0.0 ef	0.28 ± 0.0 bc
		20	37.9 ± 1.1 a	2.8 ± 0.2 b	2.45 ± 0.1 bc	0.27 ± 0.0 bc
F-test ^x^	Photoperiod		***	NS	***	NS
	Quality (A)		**	***	*	*
	Intensity (B)		**	NS	*	NS
	A × B		**	NS	**	*

^z^ Refer to Figure 1 for details on the NIL quality. ^y^ Mean (n = 3) separation within columns by Duncan’s multiple range test at 5% level. ^x^ NS, *, **, ***: Non-significant or significant at *p* ≤ 0.01.

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
