# Peer review of "Flowering and Morphogenesis of Kalanchoe in Response to Quality and Intensity of Night Interruption Light"

_plants, 2019, doi:10.3390/plants8040090_

Round 1
Reviewer 1 Report
The manuscript written by Kang et al described the effect of night interruption light (light quality and light intensity) on flowering and morphogenesis of two different cultivars of Kalanchoe. The authors used blue, red, white and white+blue lights (4h) for the interruption of long night and examined horticultural traits of the plants such as plant height, leaf length, leaf width and so on of the two cultivars. Even though the authors displayed the experimental results with Figures and tables, I could not find any scientific discoveries/logics in the manuscript (just simple description like an experimental report).
The final results are also narrow in my point of view as a research article: light quality and intensity during the night interruption affect differently morphogenesis and flowering of the two different cultivars of Kalanchoe.
1. Intensive English editing is strongly required through the whole manuscript.
2. Plant length? Does it mean “plant height”?
3. For night interruption, how many days of NI were treated?
4. Please describe why the authors treated 4h of NI? Is there any particular reason to use 4h? Usually, short time of night break has been used for flowering studies.
5. Please describe the reason why the authors added “Blue+White” as the NIL?
6. Please describe more exactly and carefully: i.e. Line 144-145: in the Figure 2A, I doubt whether we can say “the higher the light intensity, the shorter the plant length” In addition, explain more carefully figures. For example, in the Figure 2, there is no explanation about a, b and c.
7. In general, significant improvement is necessarily required for the manuscript as a research article: exact and detailed explanation, careful marking and explanation in each figure, merging scientific logics to the experimental procedure and the interpretation of experimental results.
Author Response
The manuscript written by Kang et al described the effect of night interruption light (light quality and light intensity) on flowering and morphogenesis of two different cultivars of Kalanchoe. The authors used blue, red, white and white+blue lights (4h) for the interruption of long night and examined horticultural traits of the plants such as plant height, leaf length, leaf width and so on of the two cultivars. Even though the authors displayed the experimental results with Figures and tables, I could not find any scientific discoveries/logics in the manuscript (just simple description like an experimental report).
The final results are also narrow in my point of view as a research article: light quality and intensity during the night interruption affect differently morphogenesis and flowering of the two different cultivars of Kalanchoe.
1. Intensive English editing is strongly required through the whole manuscript.
Response 1: Thank you for your review and suggestion. The manuscript was revised by a native speaker before submission and it was revised once again.
2. Plant length? Does it mean “plant height”?
Response 2: All authors agreed with the change of ‘plant length’ to ‘plant height’.
3. For night interruption, how many days of NI were treated?
Response 3: It’s already described on 2.3.1.
4. Please describe why the authors treated 4h of NI? Is there any particular reason to use 4h? Usually, short time of night break has been used for flowering studies.
Response 4: The 4h is a normal length of NI in our latitude around the world. It was the length of NI we used in previous studies also. A sentence was added to describe it in lines 34 to 35.
5. Please describe the reason why the authors added “Blue+White” as the NIL?
Response 5: We added the reason in lines 49 to 51.
6. Please describe more exactly and carefully: i.e. Line 144-145: in the Figure 2A, I doubt whether we can say “the higher the light intensity, the shorter the plant length” In addition, explain more carefully figures. For example, in the Figure 2, there is no explanation about a, b and c.
Response 6: We revised lines 128 to 129.
7. In general, significant improvement is necessarily required for the manuscript as a research article: exact and detailed explanation, careful marking and explanation in each figure, merging scientific logics to the experimental procedure and the interpretation of experimental results.
Response 7: Title of figures was revised to add more information. Also, the conclusion was revised to explain the results more clearly.

Reviewer 2 Report
This paper is presented the effects of the quality and intensity of the night interruption light (NIL) on the flowering and morphogenesis of kalanchoe. It is basically well written with good results. The most seriously thing that I concerned is in the Material and methods part, the authors should give a clear description on the total number of plants you used in each replication. Particularly in Line 105, how many plants did you measure? For the leaf length and width, which leaf did you measure on the plants? And how many leaves did you measure? What is the total number of plants for you to calculate the percentage of flowering?
As in Line 135 and 136, you mentioned that "This experiment utilized a randomized complete block design constructed of 3 replications and 2 plants per replication." With only 2 plants per replication? How can you calculate the flowering percentage and other measurements? This is very confusing! If only 2, the number is really too small to publish as scientific report.
Numbers in Table 1 and Table 2 should be presented as mean±SE, with n=?.
I think all the "plant length" in the results should be "plant height".
Table 3 and table 4, most of them are just blank. I would suggest to simply presented the results in the text, instead of such a large table.
Line 263, The conclusion need to be improved in a very clear form. please rewrite this part.
And the abstract is also need to be improved in a clearer form. Especially Line 20 to 22, it is better to present a specific result for the reader, not just using the words like "were affected", you should at least let the readers know what kind of effects, positive or negative.

Author Response
This paper is presented the effects of the quality and intensity of the night interruption light (NIL) on the flowering and morphogenesis of kalanchoe. It is basically well written with good results. The most seriously thing that I concerned is in the Material and methods part, the authors should give a clear description on the total number of plants you used in each replication.
Particularly in Line 105, how many plants did you measure? For the leaf length and width, which leaf did you measure on the plants? And how many leaves did you measure? What is the total number of plants for you to calculate the percentage of flowering?
Thank you for your review and suggestion. The number of plants used for measurement of growth parameters was already mentioned on lines 119 and 120 (total 9 plants). And ‘flowering percentage’ was also calculated using same plants.
As in Line 135 and 136, you mentioned that "This experiment utilized a randomized complete block design constructed of 3 replications and 2 plants per replication." With only 2 plants per replication? How can you calculate the flowering percentage and other measurements? This is very confusing! If only 2, the number is really too small to publish as scientific report.
The authors miscalculate the number of plants, and it was 3 plants per replication. Also, the purpose of this experiment was to examine the effects of the light quality and intensity of NIL (Night Interruption Light) on the morphogenesis and flowering of Kalanchoe, which is a short-day plant. Since the experiment was carried out in a controlled environment and also, we used vegetatively propagated clonal plants, we do not think to work many plants. In addition, the space in facility used for this experiment (the close-type plant factory, on lines 62 to 63) is limited.
Numbers in Table 1 and Table 2 should be presented as mean±SE, with n=?
Your suggestion was accepted and Tables 1 and 2 were revised.
I think all the "plant length" in the results should be "plant height".
All authors agreed to change to plant height.
Table 3 and table 4, most of them are just blank. I would suggest to simply presented the results in the text, instead of such a large table.
The data were already presented in the text in section 3.2. (line 171s, 176 and 177) and we deleted the tables.
Line 263, The conclusion need to be improved in a very clear form. please rewrite this part.
The conclusion was revised in lines 235 to 238.
And the abstract is also need to be improved in a clearer form. Especially Line 20 to 22, it is better to present a specific result for the reader, not just using the words like "were affected", you should at least let the readers know what kind of effects, positive or negative.
The abstract was revised in lines 17 to 19.

Round 2
Reviewer 1 Report
Although the authors improved the manuscript - especially in the data presentation and method description - English editing is required, in particular, in the Discussion section (i.e. Lines 210-211). Also, the same sentences are repeated in the Discussion section (i.e. Lines 204-205 and 208-209). I think that the authors should reinforce the Discussion section with scientific logic and provide plausible experimental plans, which can be done in the future.
[Minor]
In line 218, I think 'photoreceptors' looks better than 'phototropins'
In line 221, Please make it clear whether it is a gene or mutant: phyB. The Arabidopsis mutant, phyB shows early flowering, for example.
Author Response
Although the authors improved the manuscript - especially in the data presentation and method description - English editing is required, in particular, in the Discussion section (i.e. Lines 210-211). Also, the same sentences are repeated in the Discussion section (i.e. Lines 204-205 and 208-209). I think that the authors should reinforce the Discussion section with scientific logic and provide plausible experimental plans, which can be done in the future.
Thank you for your review and suggestion. The lines 204 to 205 was deleted. And we revised lines 210 to 211.
[Minor]
In line 218, I think 'photoreceptors' looks better than 'phototropins'
All authors agreed with the change of ‘phototropins’ to ‘photoreceptors’ in line 218.
In line 221, Please make it clear whether it is a gene or mutant: phyB. The Arabidopsis mutant, phyB shows early flowering, for example.
To make it clear, we changed reference 44.
